# CausalVE: Face Video Privacy Encryption via Causal Video Prediction

## Abstract

Advanced facial recognition technologies and recommender systems with inadequate privacy technologies and policies for facial interactions increase concerns about bioprivacy violations. With the proliferation of video and live-streaming websites, public-face video distribution and interactions pose greater privacy risks. Existing techniques typically address the risk of sensitive biometric information leakage through various privacy enhancement methods but pose a higher security risk by corrupting the information to be conveyed by the interaction data, or by leaving certain biometric features intact that allow an attacker to infer sensitive biometric information from them. To address these shortcomings, in this paper, we propose a neural network framework, CausalVE. We obtain cover images by adopting a diffusion model to achieve face swapping with face guidance and use the speech sequence features and spatiotemporal sequence features of the secret video for dynamic video inference and prediction to obtain a cover video with the same number of frames as the secret video. In addition, we hide the secret video by using reversible neural networks for video hiding so that the video can also disseminate secret data. Numerous experiments prove that our CausalVE has good security in public video dissemination and outperforms state-of-the-art methods from a qualitative, quantitative, and visual point of view.

## 1 Introduction

With the widespread adoption of smart devices and the Internet of Things (IoT), the security issues of biological face privacy are becoming increasingly unavoidable. The explosion of public face video distribution for IoT, exemplified by YouTube, TikTok, and Instagram, makes it difficult to protect face privacy during video interaction and distribution. In addition, the autonomy of public face video distribution and interaction on video websites means that disguised face videos must convey the same visual video information effect as the original video and hide sensitive personal privacy information.

Current face privacy measures mainly focus on destroying or hiding facial attributes. In video sequences, face attributes are destroyed by replacing the region where the person is located with blank information (Newton et al., 2005; Meden et al., 2018) or by blurring and pixellating face attributes from the detector (Sarwar et al., 2018). These methods directly damage the biometric features in facial videos, destroying the usability of data interactions and even failing to leave any useful information in interactions and propagation. To strike a balance between privacy preservation and information extraction, (Newton et al., 2005; Brkic et al., 2017; Chhabra et al., 2018; Sim & Zhang, 2015) preserve sensitive information by identifying and selectively deleting, replacing, or hiding sensitive information while preserving non-private information. The definition of sensitive information and the attacker's ability to infer hidden personal attributes from non-private information (Hu & Song, 2024) make the feasibility of these methods in preserving privacy in real-world interactions challenging. In this case, we believe that a better way to protect privacy is to perform direct and complete data hiding. One of the ways of data hiding is steganography.

The purpose of steganography is to encode sensitive information in some transmission medium and communicate covertly with a recipient who has a key to recover the secret information. zhang et al. (Zhang & Wang, 2004) proposed an image steganography method based on the human visual system, which utilizes a multi-base symbol system to dynamically adjust the embedding strength to ensure high imperceptibility. However, nowadays digital video is gradually replacing images as the main communication medium (Social), and video has a greater capacity to carry secret information, so video steganography is more necessary for development nowadays. Video steganography is a technique to embed information into the cover content. Weng et al. (Weng et al., 2019a) proposed a novel high-capacity convolutional video steganography model, which can hide a complete video clip in a video. Zhai et al. (Zhai et al., 2019) proposed a new 12-dimensional universal feature set, which is capable of detecting video steganography in a variety of embedded domains. However, in the public channel, the cover video needs to have the function of disseminating the original information, while the cover video of steganography is usually chosen randomly, and the duration and

information conveyed are not matched with the original video. How to match the cover face video with the information that needs to be disseminated by the secret video is the difficulty of applying steganographic methods such as video hiding to public video interactions.

With the great success of diffusion modeling in the field of image generation, video generation techniques have also come into the limelight. Ruan et al. (Ruan et al., 2023) proposed a joint MM-Diffusion audio-video generation framework consisting of a sequential multimodal U-Net for designing a joint denoising process. In the area of video prediction and causal inference, researchers have worked on developing advanced algorithms and techniques for accurate prediction of video content and causal analysis. Ye et al. (Ye & Bilodeau, 2023) proposed a new and efficient Transformer block for video feature learning, which reduces the complexity of a standard Transformer complexity. Hu et al. (Hu et al., 2023) proposed a Dynamic Multi-scale Voxel Flow Network (DMVFN) that uses RGB images to achieve better video prediction performance at lower computational cost. Causal inference, on the other hand, helps researchers to gain a deeper understanding of causal relationships in videos in order to better control where and how the secret information is embedded. Zang et al. (Zang et al., 2023) investigate the structure of relationships from the perspective of causal representation of multimodal data, and propose a novel inference framework for Video Question and Answer (VideoQA). Li et al. (Li et al., 2023) proposes a Context-Aware Video Intent Reasoning Model (CaVIR) to address the special VideoQA task of video intent reasoning. These studies give us new research directions for conducting privacy-secure propagation of public face videos. For a more detailed explanation of the principles, we explain the related work on video Steganography, video prediction, and face generation in appendix A.

Therefore, we introduce "CausalVE," an innovative framework for face-video privacy interaction, which significantly advances the field of video steganography and privacy protection. The novel approach integrates dynamic causal reasoning with reversible neural networks to seamlessly blend the original video content with generated cover face videos. This not only effectively conceals the identity and sensitive information within videos but also ensures that the authenticity and expressiveness of the facial features are maintained across the video timeline. The primary contributions are:

- Introduction of Dynamic Causal Reasoning for Video Prediction: The use of causal reasoning to guide the video prediction process helps in creating cover videos that are not only visually convincing but also capable of carrying hidden information without detectable alterations.

- Reversible Neural Network for Video Hiding and Recovery: The framework uses a reversible neural network that allows for the original video to be hidden within a pseudo video and accurately recovered using a key. This method provides a robust way to secure personal data while still allowing the video to be used in public channels.

- Hybrid Diffusion Model for Face Swapping: By incorporating a hybrid diffusion model that uses identity features and controlled noise processes, the system generates high-fidelity facial transformations that preserve the natural dynamics and expressions of the original video.

## 2 METHODOLOGY

### 2.1 CAUSALVE: A FRAMEWORK FOR FACE VIDEO PRIVACY INTERACTION

Figure 1 shows the specific framework of our CausalVE. For a given face video, we first extract the first frame for face replacement. Then, the physical information of the original video is used as a guide and the causal analysis framework is applied to build the time series of the overlay video Rai for video prediction. Finally, the overlay video is generated. The original video is hidden in the overlay video by a reversible neural network to generate a pseudo-video with the information of the original video. Anyone can view the pseudo video directly and the key holder can restore the pseudo video to the original video by using the key.

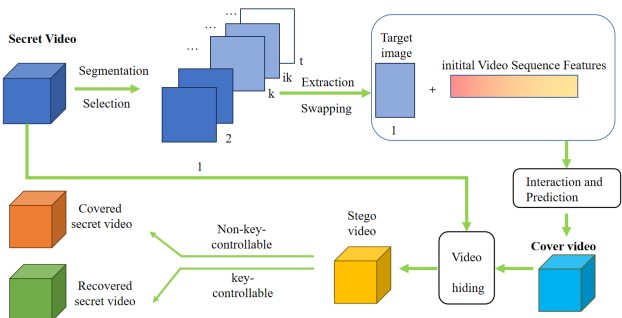

Figure 1: CausalVE Network Framewrok

### 2.2 COVER VIDEO GENERATION

Since the cover face video is not only for the attacker but also for the public channel at the same time, the generation of the cover face video needs to consider not only the detection of counterfeiting tools but also the visual representation of the face. Therefore, we propose the use of dynamic causal reasoning to support video prediction for the generation of cover-face videos. In the following, we will elaborate on the network module for cover face video generation.

### 2.2.1 FACE SWAPPING MODULE

For a video space $\mathbb{R}^{3 \times H \times W \times T}$, we divide into n video frames by frequency. Take the first frame image as $I_s$ for face-swapping. In order to make the transformed face have better security on this graph space, and at the same time make the transformed face connect with the post-time series better, we draw on the idea of the diffusion model to perform image face-swapping.

Consider $I_s$ and $I_t$ to represent the secret and target images, respectively, each containing distinct facial features $F_s$ and $F_t$. The primary goal of this research is to construct a cover image $I_b$, wherein $F_t$ is seamlessly replaced by $F_s$, while meticulously preserving the identical pose and expression.

Let $I$ denote an image within the space $\mathbb{R}^{3 \times H \times W}$. For identity embedding, we employ the ArcFace model (Deng et al., 2019). Upon embedding the secret image $I_s$, we obtain the identity feature $I_{id}$. This feature $I_{id}$ is then integrated into the diffusion model $\lambda_\theta(x_t, t, I_{id})$. At each time step $t$, $I_s$ is subject to a prior process that aims to reconstruct $I_t$ utilizing a standard Gaussian distribution (Luo, 2022), which is achieved by reversing the recursive noise addition, as defined in the following equation:

$$q\left(x_t | x_{t-1}\right) = \mathcal{N}\left(x_t; \sqrt{1 - \beta_t} x_{t-1}, \beta_t I\right) \tag{1}$$

where $\beta_t$ is a predefined variance schedule. The inverse diffusion process is described by:

$$p_\theta\left(x_{t-1} | x_t\right) = \mathcal{N}\left(x_{t-1}; \mu_\theta(x_t, t), \sigma_\theta(x_t, t)\right) \tag{2}$$

To facilitate the generation of the cover image, multiple expert models are utilized to provide nuanced facial guidance, thus enhancing the fidelity of the synthesized image. The incorporation of multiple models often introduces various forms of noise, complicating the retention of the target background during the face-swapping process. To mitigate this, we introduce a novel target-preserving hybrid method that modulates the mask's strength by gradually increasing its intensity from 0 to 1 over the diffusion process duration $T$. This modulation is strategically controlled to ensure the preservation of the target image's structural integrity, as expressed by:

$$U_t = \min\left\{1, \frac{T - t}{\hat{T}} U\right\} \tag{3}$$

Here, $U$ denotes the rigid mask derived through the face parsing process, and $U_t$ represents the dynamic mask, which increases in intensity progressively. The threshold $\hat{T}$ defines the critical point at which the mask transitions to its full intensity.

The blending of the intermediate predictions $\hat{x}_{t-1}$ with the target images is then performed using the mask $U_t$ in the reverse process:

$$\hat{x}_{t-1} = (1 - U_t) \cdot \varphi_\theta(x_t, t) + U_t \cdot \lambda_\theta(x_t, t, I_{id}) \tag{4}$$

This process effectively allows the integration of facial features from $I_s$ into $I_t$, culminating in the creation of the desired cover image $I_b$, which maintains the original pose and expression of the target while featuring the secret identity.

### 2.2.2 INTERACTION MODULE

The original video contains rich temporal, spatial, and physical information. To utilize this information for assisting subsequent dynamic video reasoning, we design an interaction model to incorporate various desirable types of inputs. As shown in Figure 2, this module fuses the audio sequence information with the currently selected face-changing picture frame to generate a new representation and updates the fusion process by simulating facial head movement through conditional VAE for cover video prediction.

Specifically, we set the first frame of the secret video to the face-swapping picture $I_s$. At this time, in the 3D deformable model (3DMM), the face shape can be expressed by the following formula:

$$F = \bar{F} + \alpha r_{id} + \beta r_{ex} \tag{5}$$

Here, F represents the average shape of the 3D face, rid represents the orthogonal basis for shape, and rex represents the orthogonal basis for expression, with $\alpha$ and $\beta$ describing human identity and expression, respectively. To maintain posture changes, the coefficients $\mu$ and $\nu$ represent head rotation and transformation, respectively. To separate these parameters from the human body, we incorporate audio modeling parameters $\beta$, $\mu$, and $\nu$. The partial posture parameters of the head, denoted as $\rho = [\mu, \nu]$, are used for personal identity ID inquiries. We establish the correlation between expression and the identity of a specific individual through the expression coefficient $\beta_0$ of $I_t$. To reduce the weight of expressions of other facial components when speaking, the lip motion coefficients generated by pre-trained Wav2lip are used as targets. In addition, other micro-expressions are constrained by additional key point losses. Then, the expression coefficient of t frames is generated through the audio sequence, where the audio feature of each frame is the 0.2s Mel spectrum. We utilize a ResNeXt-based audio encoder $\Psi$ to map to a latent space, and then the residual layer acts as a

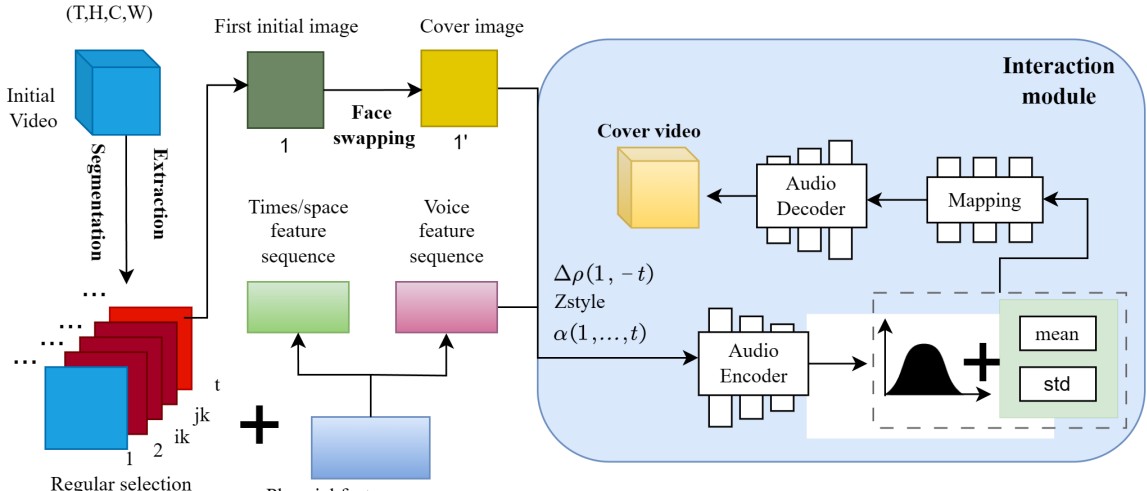

Figure 2: Architecture of CausalVE for Cover Video Generation. After segmenting the initial video, CausalVE selects images for face-swapping at regular intervals. The Interaction Module then uses a single cover image to generate a complete cover video sequence, guided by the voice sequence from the initial video.

mapping network $\Theta$ to decode the expression coefficients. To maintain the facial expression features of the cover video, we introduce the reference label $\beta_0$ to control personal features, and only use the lip area as the ground truth during training. Finally, the framework of this network can be expressed as:

$$\beta_{\{1,\dots,t\}} = \Theta\left(\Psi\left(\alpha_{\{1,\dots,t\}}\right), \beta_0 \odot \bar{F}\right) \tag{6}$$

Where $\odot$ is the element-wise product, which represents the fusion process of the representation of the replacement picture and the original video sequence.

To produce extended, consistent, and uninterrupted head movements. We made some changes to the above framework. We changed the encoder part to the architecture of implementing ResNeXt with VAE's encoder, at this time, the framework of this network not only learns the potential distribution of the input data but also applies ResNeXt can help the model to capture more complex image features. In addition, according to the idea of CVAE, we also add the corresponding head motion parameter $\rho = [\mu, \nu]$, the style coefficient $z_{style}$ as an added input, which makes the model pay more attention to the rhythm and personal style, embedded with a Gaussian distribution, and the distribution and quality of the generated motion is measured by $L_1$, and the decoder network learns based on the distribution sampled to generates a gestalt map with the same number of frames as the audio sequence.

### 2.2.3 RE-PREDICTION AND DECISION MODULE

During the interaction between the target image and the original video voice sequence, we use voice-controlled video generation to obtain the cover video. However, in such a prediction process, the cover video controls the expression and character characteristics through $\beta_0$ and controls the character style through the specified function $z_{style}$. These controls are highly subjective and dependent, completely interacting with $I_t$ as the core representation. This method can have good results when the distance to $I_t$ is closer to the number of frames. However, the representation of $I_t$ becomes less and less special as the time series increases. At this time, as the number of frames $t_s$ continues to increase, the representation of $I_t$ becomes less and less obvious in actual situations. However, in the process of using voice interaction prediction, $I_t$ is still regarded as the core interaction, and the information conveyed by the cover video would be contrary to the facts. Our goal is to conceal sensitive personal information during face video interactions on public channels while ensuring effective interaction and dissemination of other information. To solve the above Questions, we perform video re-prediction and decision-making through causal video prediction.

We re-examine the entire incident. When generating a cover face video, we need to achieve the following goals:

**Video character event prediction.** This task requires a model to infer possible future facial biometric changes based on observed videos.

**Reverse reasoning.** The task is to guess the facial biometric changes that occurred before the start of an existing video clip.

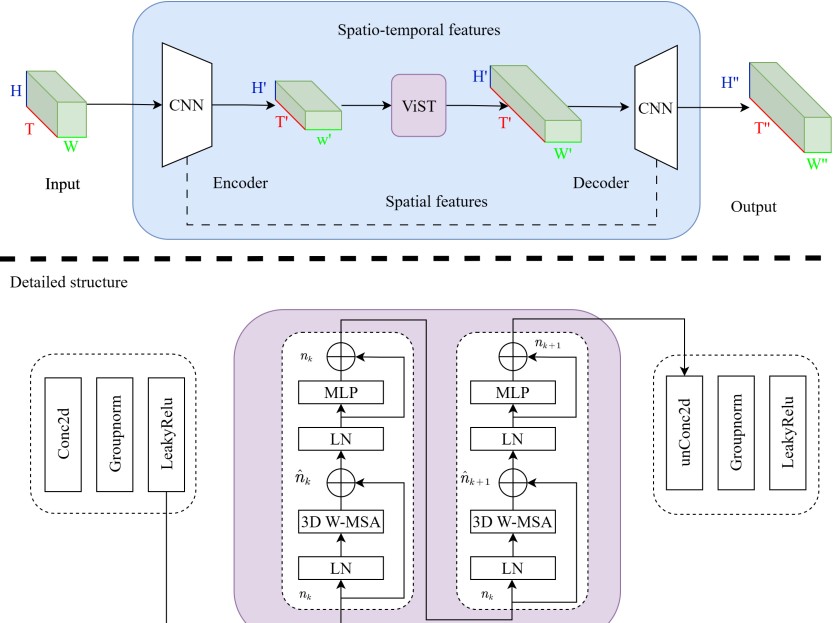

Figure 3: The CNN-ViST-CNN Frameworks for Video Prediction in our Re-prediction and Decision Module. We utilize CNNs as the encoder for extracting spatial features and as the decoder for post-video frame prediction. And Video Swin Transformer (ViST) is employed as the translator to learn both temporal and spatial evolution.

**Counterfactual reasoning.** This task guesses the inevitable results given some assumptions (for example, what will happen if you don't smile?) The hypothetical conditions will not appear in the original video that requires reasoning, so the model needs to imagine and reason Video prediction results under some given assumptions.

In this process, if our use of $I_t$ for face video prediction violates any of the above reasoning processes, we need to use other physical information of the original video sequence to control the prediction of subsequent videos. We introduce the Video Swin Transformer (ViST) to perform dynamic face video inference and prediction based on the latent spatial dynamics of the original video. Figure 3 shows the frameworks of our CNN-ViST-CNN Video Prediction framework. In this network framework, we adopt CNN as the encoder for extracting spatial features and the decoder for post-video frame prediction, and the Video Swin Transformer (ViST) as the translator for learning temporal and spatial evolution.

The encoder stacks $n_i$ Conv2d, LayerNorm, and ReLU for convolution, which is expressed as follows:

$$\Omega_k = \sigma\left(LN\left(C\left(\Omega_{k-1}\right)\right)\right) \tag{7}$$

Where LN represents LayerNorm and C represents Conv2d. The input $\Omega_{k-1}$ and the output $\Omega_k$ shapes are $(T_{K-1}, C_{K-1}, H_{K-1}, W_{K-1})$ and $T_k, C_k, H_k, W_k, 0 < k < n_i$. The decoder and encoder use the same number of layers for decoding and prediction.

We use Video Swin Transformer as a tool for spatiotemporal evolution analysis. The core principle is to use the Transformer architecture to simultaneously process the spatial characteristics and temporal continuity of video data. By limiting the calculation of 3D self-attention to a local window, the computational complexity can be significantly reduced. The shift window strategy changes the arrangement of windows between consecutive layers, promotes information exchange between different windows, and enhances the overall perception of the model. The formula is expressed as follows:

$$\text{Attention}(Q, K, V) = \text{Softmax}\left(\frac{QK^T}{\sqrt{d_k}}\right) V \tag{8}$$

Where $Q, K$, and $V$ are query, key and value matrices respectively. They may originate from different parts of the video frame in the same window, or the same position at different time points. $d_k$ is the dimension of the key vector.

By adopting a hierarchical architecture, features at different scales are captured by gradually reducing the temporal and spatial resolution. This approach helps the model capture extensive contextual information while maintaining low computational cost. The specific expression formula is:

$$\text{MultiHead}(Q, K, V) = \text{Concat}(\text{head}_1, \ldots, \text{head}_h)W^O \tag{9}$$

$$\text{head}_i = \text{Attention}(QW_i^Q, KW_i^K, VW_i^V) \tag{10}$$

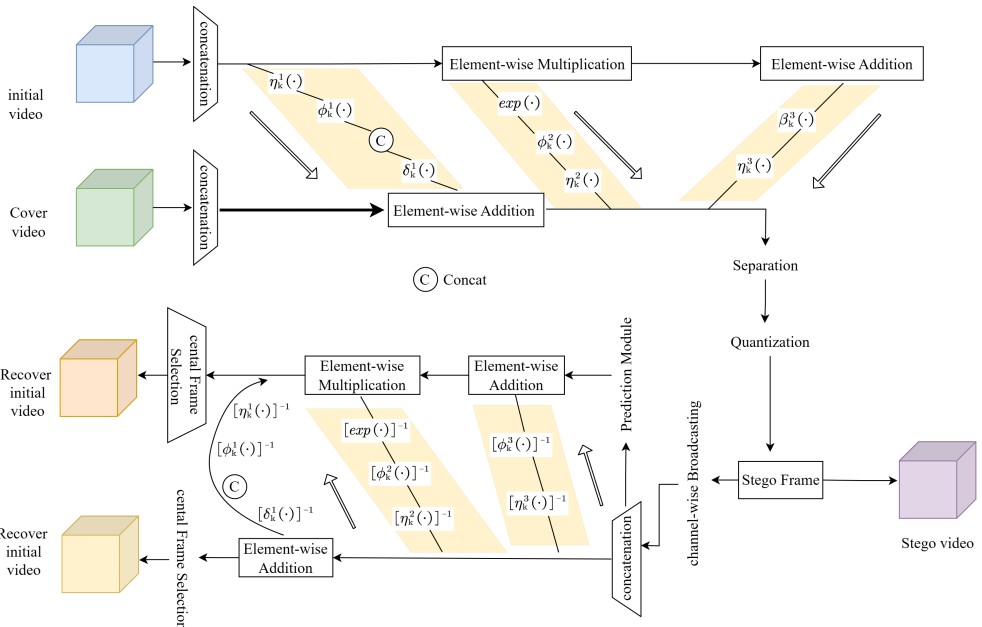

Figure 4: Video Hiding and Recovery Framework. Given a hidden video $x_{secret}$ and a cover video $x_{cover}$ after forward hiding frame by frame, a pseudo-video $x_{stego}$ is generated. Inversely, with the same reversible neural network and same parameters, the pseudo-video $x_{stego}$ can be recovered into the original video $x_{recover}$.

$W_i^Q$, $W_i^K$, and $W_i^V$ are the linear transformation weights of the corresponding heads, and $W^O$ is the linear transformation weight of the output.

## 2.3 VIDEO HIDING AND RECOVERY

Figure 4 shows the framework of the hidden part of our video. Specifically, given a hidden video $x_{secret}$ and a cover video $x_{cover}$ (the cover video is generated from the above section) after forward hiding frame by frame, a pseudo-video $x_{stego}$ is generated, which is ostensibly indistinguishable from $x_{cover}$ to achieve the $x_{secret}$ undetectable the effect of $x_{secret}$. With the same reversible neural network architecture and parameters, the pseudo-video $x_{stego}$ can be recovered into the original video $x_{recover}$. In order to utilize the temporal and spatial correlation within the video, we use Discrete Wavelet Transform (DWT) to divide each frame into four frequency bands LL,HL,LH,HH, and then in the same group of frames, we connect the portions of the same band portion of different frames in the channel dimension, and then concatenate these four bands in series according to the frequency magnitude, to generate the final secret video for concealment $x'_{secret}$ and the cover video $x'_{cover}$.

It can be shown in Figure 4 that our video hiding and recovery results in reversible video hiding by constructing the reverse information flow through the invertible block. The initial reversible neural network can be defined as follows: assuming the input is x,the hiding module splits x into two parts $x_1$ and $x_2$ along the channel axis by the transform parameter and then untransforms it by the same transform parameter (Dinh et al., 2014), which is expressed as follows:

$$\hat{\mathbf{x}}_1 = \mathbf{x}_1 \cdot \gamma_1(\mathbf{x}_2) + \varepsilon_1(\mathbf{x}_2)$$
$$\hat{\mathbf{x}}_2 = \mathbf{x}_2 \cdot \gamma_2(\hat{\mathbf{x}}_1) + \varepsilon_2(\hat{\mathbf{x}}_1), \tag{11}$$

Where $\gamma(\cdot)$ and $\varepsilon(\cdot)$ are the functions of the invertible block transformation.

On this basis, we construct reversible projections in the inversion block by means of several interaction paths between the two branches: using additive transformations to project $x'_{cover}$ and multiplicative transformations to project $x'_{secret}$, and generating the transformation parameters from each other. Here, we utilize the weighting modules ($\eta_k^1(\cdot)$ and $\phi_k^1(\cdot)$) to extract features from all the secret sets, producing a feature set. $\eta_k^i(\cdot)$ and $\phi_k^i(\cdot)(i = 1, 2, 3)$ refer to a $3 \times 3$ convolutional layer and a five-layer dense block, respectively. The transformation parameters of $x_{secret}$ can be generated by $x'_{cover}$, so that the bijection of the front propagation of the video hiding is reformulated as in the hidden module:

$$\mathbf{x}'^{k+1}_{cover} = \mathbf{x}'^k_{cover} + \xi_k\left(||\phi_k^1\left(\eta_k^1\left(\mathbf{x}'^k_{secret}\right)\right)||\right)$$
$$\mathbf{x}'^{k+1}_{secret} = \mathbf{x}'^k_{secret} \cdot \exp\left(\phi_k^2\left(\eta_k^2\left(\mathbf{x}'^{k+1}_{cover}\right)\right)\right) + \phi_k^3\left(\eta_k^3\left(\mathbf{x}'^{k+1}_{cover}\right)\right), \tag{12}$$

where $||\cdot||$ refers to the channel-wise concatenation. $\exp(\cdot)$ is the Exponential function. Accordingly, the backward recovery is expressed as follows:

$$\mathbf{X'}_{secret}^{k} = \left(\mathbf{X'}_{secret}^{k+1} - \phi_k^3\left(\eta_k^3\left(\mathbf{X'}_{cover}^{k+1}\right)\right)\right) \cdot \exp\left(-\phi_k^2\left(\eta_k^2\left(\mathbf{X'}_{cover}^{k+1}\right)\right)\right)$$
$$\mathbf{X'}_{cover}^{k} = \mathbf{X'}_{cover}^{k+1} - \xi_k\left(||\phi_k^1\left(\eta_k^1\left(\mathbf{X'}_{\sec ret}^{k}\right)\right)||\right). \tag{13}$$

## 2.4 Loss Function

Our CausalVE loss function consists of cover image generation loss, initial cover video generation loss, video prediction loss, causal inference and decision loss, and video hiding loss. We provide additional information about the loss function in the appendix B for more supplementary information.

## 3 Experiments

### 3.1 Experimental Setup

**Datasets and Settings.** The VoxCeleb2 (Chung et al., 2018) training set is used to train our CausalVE, with the spatial resolution of each sequence contained in it fixed to $512 \times 300$. During training, we randomly crop the training video to $256 \times 256$ and randomly flip it horizontally and vertically to increase the amount of data. The testing datasets include VoxCeleb2, with 150,480 videos at each sequence resolution $512 \times 300$, Voxblink (Lin et al., 2024) with 1.45 million videos by about 38000 people at each sequence resolution $480 \times 367$, and Mead (Wang et al., 2020). We segment the video in Mead and get 54291 videos by about 60 people at each sequence resolution $256 \times 256$. The test video for each sequence uses a center crop to $256 \times 256$ to ensure that the cover video and the secret video have the same resolution. The optimizer uses Adam's standard parameters, while the initial learning rate is $1 \times 10^{-5}$, halved every 25K iterations. An NVIDIA A100 Tensor Core GPU is used for all training and testing.

**Benchmarks and Evaluation Metrics.** We evaluate the soundness of our motivation and the effectiveness of our CausalVE. We compare our CausalVE with different information steganography approaches, including LSB, Weng et al. (Weng et al., 2019b), Baluja et al. (Baluja, 2020), HiNet (Jing et al., 2021), RIIS (Xu et al., 2022), and LF-VSN (Mou et al., 2023b). It is important to note that the original video-hiding model was initially designed solely for information concealment, differing from our configuration for face privacy interaction protection. To accommodate video concealment, we made slight modifications to the output dimensions. Furthermore, unlike our approach where the model autonomously generates cover videos, the cover videos for these models were predefined. To align these models with our methodology, we adjusted the cover video inputs, redefined the video generation function, and retrained the networks. We use two metrics to evaluate the quality of cover/hide and secret/reduce video pairs, namely cover/stego and secret/recovery video peak signal-to-noise ratio (PSNR), and structural similarity index (SSIM) (Wang et al., 2004), MAE and RMSE.

Meanwhile, in order to verify the validity of the generated overlay images, we use ArcFace (Deng et al., 2019), CosFace (Wang et al., 2018a), SphereFace (Liu et al., 2017a) and AdaFace (Kim et al., 2022b) face recognition models to verify the processing effect of RFIS-FPI on cover images, recovery images. Since the resolution of ArcFace (Deng et al., 2019) and AdaFace (Kim et al., 2022b) is $112 \times 112$, CosFace (Wang et al., 2018a) and SphereFace (Liu et al., 2017a) have a resolution of $112 \times 96$, which is smaller than the resolution of our training dataset, we use MTCNN (Xiang & Zhu, 2017) to align and crop the face images to match the resolution of RFIS-FPI. We use SSIM (Wang et al., 2004) and Learning to Perceive Image Patch Similarity (LPIPS) to perceive the quality of cover/secret and secret/recovery image pairs. For SSIM / LPIPS values between secret and cover images, we denote them by $\text{SSIM}_{st}$ / $\text{LPIPS}_{st}$. Similarly, for the SSIM / LPIPS value between the secret image and the recovery image, we denote it by $\text{SSIM}_{sr}$ / $\text{LPIPS}_{sr}$. A higher SSIM means that the two images are more similar, and a lower LPIPS means that the two images are less similar.

Moreover, we use the statistical steganalysis tool StegExpose (Boehm, 2014) to evaluate the effectiveness of our face privacy interaction protection approach.

### 3.2 Quantitative Comparison

Tables 1 and 2 demonstrate the effectiveness of our method and other state-of-the-art video hiding methods on video datasets. The best data in each column of the table is in red, and the second best data is in blue. As can be seen from Table 1, our CausalVE compares slightly better metrics than other video hiding methods on cover video and steganography video. This is due to the fact that excellent generated videos have a camouflage ability that is no less than that of natural videos. At the same time, the cover video with the same ability to evolve time sequence and spatial features is similar in the process of hidden reorganization of reversible neural networks. This makes it harder to spot the difference between a fake video and a cover video. In Table 2, the performance of our method on secret images/recovered images is not far from the results of the state-of-the-art methods. This shows that our optimized reversible neural network applied to face video encryption is effective. The use of CausalVE allows for public face video video information interaction while having the perfect role of carrying sensitive information interaction.

To verify the effectiveness of our face cover video generation method, we compare our CausalDE with specialized face video generation methods and biometric privacy generation methods, and the results are shown in Table 3. Our method performs well in generating cover face videos, fully hiding the face information while achieving a more complete communication of the original video information.

Table 1: **Benchmark comparisons about Cover/Stego video pair**

| Methods | Cover/Stego video | | | | | | | | | | | |
|---|---|---|---|---|---|---|---|---|---|---|---|---|
| | VoxCeleb2 | | | | Voxblink | | | | Mead | | | |
| | PSNR(dB)↑ | SSIM↑ | MAE↓ | RMSE↓ | PSNR(dB)↑ | SSIM↑ | MAE↓ | RMSE↓ | PSNR(dB)↑ | SSIM↑ | MAE↓ | RMSE↓ |
| 4bit-LSB | 33.30 | 0.689 | 6.84 | 7.94 | 33.28 | 0.723 | 7.29 | 9.13 | 33.66 | 0.741 | 6.42 | 8.40 |
| Weng et al. | 39.77 | 0.851 | 3.24 | 4.87 | 38.90 | 0.884 | 3.99 | 3.92 | 37.36 | 0.853 | 4.73 | 5.26 |
| Baluja et al. | 36.74 | 0.965 | 3.78 | 5.02 | 38.39 | 0.854 | 3.73 | 7.42 | 38.59 | 0.865 | 4.15 | 5.43 |
| HiNet | 37.23 | 0.969 | 2.94 | 3.45 | 40.70 | 0.946 | 3.36 | 4.11 | 43.80 | 0.937 | 3.61 | 4.32 |
| RIIS | 45.20 | 0.967 | 3.17 | 3.41 | 46.23 | 0.964 | 3.43 | 3.95 | 44.79 | 0.939 | 3.82 | 4.13 |
| LF-VSN | 47.21 | 0.968 | 2.63 | 2.82 | 49.72 | 0.983 | 2.94 | 3.75 | 48.71 | 0.959 | 3.23 | 3.71 |
| CausalVE | 50.39 | 0.975 | 2.65 | 2.73 | 50.74 | 0.989 | 2.79 | 3.62 | 50.72 | 0.972 | 3.10 | 3.32 |

Table 2: **Benchmark comparisons about Secret/Recovery video pair**

| Methods | Secret/Recovery video pair | | | | | | | | | | | |
|---|---|---|---|---|---|---|---|---|---|---|---|---|
| | VoxCeleb2 | | | | Voxblink | | | | Mead | | | |
| | PSNR(dB)↑ | SSIM↑ | MAE↓ | RMSE↓ | PSNR(dB)↑ | SSIM↑ | MAE↓ | RMSE↓ | PSNR(dB)↑ | SSIM↑ | MAE↓ | RMSE↓ |
| 4bit-LSB | 24.20 | 0.695 | 6.73 | 7.85 | 33.25 | 0.648 | 7.29 | 9.13 | 33.64 | 0.643 | 6.43 | 8.40 |
| Weng et al. | 34.60 | 0.811 | 3.37 | 5.06 | 38.90 | 0.877 | 4.01 | 5.92 | 37.63 | 0.859 | 4.69 | 5.28 |
| Baluja et al. | 35.24 | 0.841 | 3.45 | 5.52 | 36.39 | 0.855 | 4.97 | 7.42 | 36.60 | 0.853 | 3.62 | 4.42 |
| HiNet | 41.70 | 0.922 | 3.19 | 4.13 | 43.73 | 0.917 | 3.58 | 4.73 | 42.73 | 0.935 | 3.12 | 4.32 |
| RIIS | 43.09 | 0.935 | 2.93 | 3.63 | 44.76 | 0.934 | 3.54 | 4.71 | 44.79 | 0.939 | 3.16 | 4.36 |
| LF-VSN | 47.87 | 0.957 | 2.97 | 3.17 | 46.72 | 0.959 | 2.56 | 3.72 | 49.73 | 0.967 | 3.13 | 4.35 |
| CausalVE | 48.15 | 0.972 | 2.88 | 3.09 | 49.72 | 0.975 | 2.48 | 3.71 | 50.76 | 0.963 | 2.64 | 3.26 |

Table 3: **Comparison of image visual quality metrics and face cosine similarity(con-sim) between different face recognition algorithms for face privacy interaction protection.**

| Methods | $SSIM_{st}$ ↓ | $LPIPS_{st}$ ↑ | $SSIM_{sr}$ ↑ | $LPIPS_{sr}$ ↓ | cos-sim |
|---|---|---|---|---|---|
| ArcFace | 0.029 | 0.942 | 0.971 | 0.091 | 0.997 |
| CosFace | 0.058 | 0.953 | 0.963 | 0.107 | 0.996 |
| SphereFace | 0.063 | 0.947 | 0.951 | 0.114 | 0.996 |
| AdaFace | 0.032 | 0.954 | 0.965 | 0.109 | 0.997 |

### 3.3 QUALITATIVE COMPARISON

Figure 5a shows a comparison of the effectiveness of our CausalVE and the LF-VSN method, which produces the next best results for hidden images, on hidden videos. As can be seen in Figure 5a, our CausalVE is closer to visual logic on hidden videos thanks to guided video prediction. Specifically, we chose a challenging task: a piece of speech with multiple intonational auxiliaries: "Mum ... Ah-Haha!", a scenario that requires challenging matching videos to hide and show the progression from a closed accent to an open accent to a toothy smile. Under the same time sequence, we extracted the same frames from the two representative methods for comparison. Our CausalVE effect is closer to the truth. The coherent start-to-open movement from "Mum" to "Ah" is more consistent with the original semantics, which makes our CausalVE visually superior to the LF-VSN.

### 3.4 STEGANOGRAPHIC ANALYSIS

Data security remains a critical issue in the field of steganography. This section evaluates the resistance of various steganographic methods to detection by steganalysis tools, focusing on their ability to differentiate stego frames from natural frames. We utilize StegExpose for this evaluation, creating a detection dataset composed of stego and cover frames in equal proportions. Detection thresholds are varied extensively within StegExpose (Boehm, 2014), and the resulting data is represented on a receiver operating characteristic (ROC) curve, shown in Figure 5b. Notably, an ideal detection scenario is where the probability of identifying stego frames from a balanced mix is 50%, akin to random chance. Thus, a ROC curve that approximates this ideal indicates higher methodological security. Our findings demonstrate that the stego frames produced by our CausalVE model are significantly more difficult to detect than those from other methods, highlighting the enhanced data security offered by our CausalVE.

Additionally, to verify the effectiveness of the CausalVE module, ablation experiments were conducted on both the causal analysis and video generation modules. These ablation experiments are shown in Appendix C.

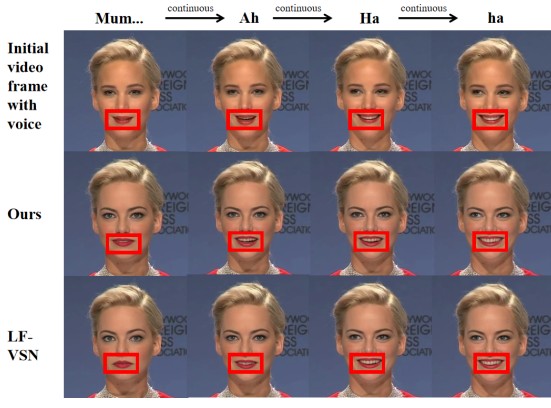

(a) After video hiding by our method and LF-VSN method for videos of the same time sequence, the resulting graphs of comparison of the hidden videos generated by different methods and the same frame of each syllable of a sentence in the original video are taken as images.

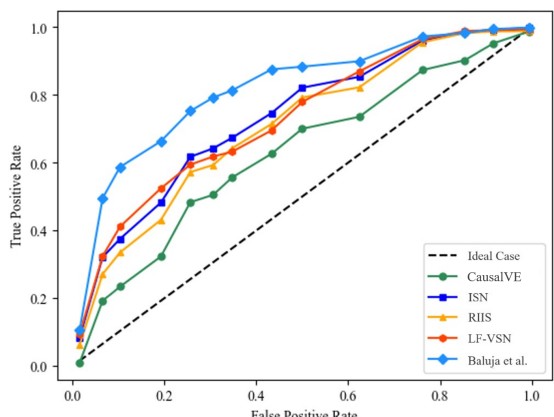

(b) Analysis of steganalysis resistance across various techniques by StegExpose (Boehm, 2014) reveals a noteworthy trend: as accuracy approaches 50%, resilience to vector analysis increases.

Figure 5: Overall comparison and steganalysis results.

## 4 CONCLUSIONS

We provide the CausalVE framework, which demonstrates a high level of efficiency in hiding and recovering video content, and positions it as a leading solution for privacy protection in video content shared over public channels. The experimental results reveal that CausalVE outperforms existing methods in both the visual quality of the cover videos and the undetectability of the hidden content, offering substantial improvements over traditional steganography and face-swapping techniques. The findings suggest that CausalVE not only provides robust privacy protection but also ensures that the integrity and expressiveness of the video content are maintained, making it a valuable tool for various applications in digital media, communications, and security fields. Furthermore, the approach's resistance to steganalysis tools underscores its potential for secure communication channels, where maintaining confidentiality and authenticity is crucial. Overall, this work lays a strong foundation for future research in video privacy protection, particularly in developing methods that balance security with the need for expressive and dynamic video content.

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

## A  RELATED WORK

### A.1  VIDEO STEGANOGRAPHY

Video steganography involves embedding a secret message within a video, in a way that is almost imperceptible to humans. The Least Significant Bit (LSB) method is a traditional steganographic technique based on spatial domain, which substituting the n least significant bits of a pixel in the video frame with the n most significant bits of the secret message. (Swathi & Jilani, 2012) applied LSB in video steganography to hide secret text in grayscale video frames. (Abbas et al., 2015) combined LSB technique with Cuckoo Search to embed each secret image's color

channel independently into a cover video's frame. Beyond spatial-domain methods, some transform-domain techniques were also applied in video steganography, such as discrete cosine transform (DCT) (Hsu & Wu, 1999) and discrete wavelet transform (DWT) (Barni et al., 2001). Transform-domain methods, though more undetectable and robust than spatial-domain methods, still provide a limited capacity for embedding secret information, ranging from text (Patel et al., 2013; Abbas et al., 2015; Mstafa et al., 2017) to images (Lu et al., 2010; Kumar & Singh, 2018; Sadek et al., 2017).

Recently, some deep learning models (Hayes & Danezis, 2017; Baluja, 2017; 2020; Guan et al., 2022) have been proposed for video steganography, achieving superior performance compared to traditional methods. (Hayes & Danezis, 2017) introduced the application of Generative Adversarial Network (GAN) in steganography, demonstrating that employing an adversarial training approach can enhance the security of concealment. (Baluja, 2017; 2020) first implemented concealing a full-sized image within another image. (Guan et al., 2022) attempts to enhance capacity by embedding multiple images within a video, bringing it closer to true video steganography. Video hiding is an important research direction of video steganography, which attempts to hide a whole video into another one. Different from above methods, it requires larger hiding capacity. (Weng et al., 2019b) was the first to explore concealing/recovering a video within/from another video by masking the residuals between consecutive frames on a frame-by-frame basis. Besides, (Mishra et al., 2019) explores temporal correlations in video steganography using 3D CNNs. (Mou et al., 2023a) further extends the capacity limit of video steganography, enabling hiding/recovering 7 secret videos in/from 1 cover video. In conclusion, the previous research demonstrate the prospects of deep learning in video hiding.

## A.2 VIDEO PREDICTION

Video prediction involves predicting future video frames based on given ones. According to their model architecture, video prediction methods can be categorized as recurrent-based and recurrent-free. Recurrent-based models process predictions by incorporating previously predicted frames into the current input, making the prediction sequence serial in nature. PredRNN (Wang et al., 2017) utilizes standard ConvLSTM (Shi et al., 2015) modules to develop a Spatio-temporal LSTM (ST-LSTM) unit that concurrently captures spatial and temporal changes. The advanced PredRNN++ (Wang et al., 2018b) introduces a gradient highway unit to address the issue of vanishing gradients and a Casual-LSTM module for cascading spatial and temporal memories. Enhancements in PredRNNv2 (Wang et al., 2023) include the introduction of a curriculum learning approach and a memory decoupling loss to enhance performance. MIM (Wang et al., 2018c) incorporates high-order non-stationarity into the design of LSTM modules. PhyDNet (Le Guen & Thome, 2020) separately models PDE dynamics and additional unknown information using a recurrent physical unit. E3DLSTM (Wang et al., 2019) merges 3D convolutions with recurrent networks, and MAU (Chang et al., 2021) features a motion-aware unit that efficiently captures motion dynamics. Despite the development of various sophisticated recurrent-based models, the underlying mechanisms contributing to their efficacy are still not fully understood.

On the other hand, recurrent-free models simplify the prediction process by inputting the entire sequence of observed frames and producing all predicted frames simultaneously. Due to its parallel characteristic, it has an inherent efficiency advantage over the recurrent-based model. Recurrent-free models often utilize 3D convolutional networks to handle temporal dependencies (Liu et al., 2017b; Aigner & Körner, 2018). Early on, PredCNN (Xu et al., 2018) and TrajectoryCNN (Liu et al., 2021) employed 2D convolutional networks to prioritize computational efficiency. Initially, these early recurrent-free models were criticized for their poor performance. However, models like SimVP (Gao et al., 2022; Tan et al., 2022; 2023) have recently demonstrated a simple yet effective approach that rivals recurrent-based models. PastNet (Wu et al., 2023) and IAM4VP (Seo et al., 2023) represent newer developments in recurrent-free models, showcasing notable improvements. In this study, we have implemented both recurrent-based and recurrent-free models within a single framework to methodically examine their intrinsic characteristics. Furthermore, we have explored the capabilities of recurrent-free models by redefining the spatio-temporal prediction challenge and integrating MetaFormers (Yu et al., 2022) to connect the visual backbone with spatio-temporal predictive learning more effectively.

## A.3 FACE SWAP MODEL

Mainstream Face swapping techniques are primarily divided into two groups: 3D-based methods and GAN-based methods. The 3D-based approaches (Blanz et al., 2004; Nirkin et al., 2017) typically utilize the 3DMM (Blanz & Vetter, 1999) to integrate structural priors. However, these methods often require human intervention or tend to produce noticeable artifacts. On the other hand, GAN-based methods generally focus on the target, merging identity features from the source face with the target's characteristics and employing GANs to maintain the authenticity of the swapped face. Nonetheless, these techniques often involve numerous loss functions, and balancing them necessitates meticulous adjustment of the hyperparameters. Additionally, these methods usually make only slight alterations to the target face, which limits their effectiveness in scenarios where there is a significant discrepancy in facial shapes between the source

and the target. While some studies have attempted to use features from the 3DMM (Li et al., 2021; Wang et al., 2021) to guide the face swapping process, this indirect use of 3D information still falls short in maintaining consistent facial shapes.

Recently, diffusion model (Ho et al., 2020; Nichol & Dhariwal, 2021; Rombach et al., 2022) has been introduced for face swapping due to its delicate controllability and high fidelity. DiffFace (Kim et al., 2022a) trained ID Conditional DDPM with facial guidance to preserve target insensitive attributes. DiffSwap (Zhao et al., 2023) designed a 3D-aware masked diffusion model using designed face attributes, enabling high-fidelity and controllable face swapping. The previous work demonstrates the great potential of the diffusion model in face swapping.

## B  LOSS FUNCTION DETAILS

### B.1  COVER IMAGE GENERATION LOSS.

Cover image generation is performed in a noisy environment, while the expert model is trained on a clean image. Therefore, we predict the noise by using the denoising score matching loss, while the identification loss due to effective recognition of faces by multi-expert recognition can be summarised by the source identification at each time step t. Specifically, the cover image generation loss is formulated as follows:

$$\mathcal{L}_{cover} = \|\sigma - \sigma_\theta\left(x_t, t, I_{id}\right)\|_2^2 + 1 - \cos\left(I_{id}, \hat{I}_{id}\right) \tag{14}$$

Where $\cos\left(-, -\right)$ denotes the cosine similarity between the un-noised secret identity and the denoised secret identity.

### B.2  INITIAL COVER VIDEO GENERATION LOSS.

In the initial cover video generation process, we use the first cover image to generate the corresponding video. This is an image-to-video process. By using the pre-trained expression coefficients obtained from wav2lip, then, 3D face capture is performed on the face of the target image as a target to guide the video generation. At this point, both the generation coefficients and the lip movement expression coefficients are known, and the face capture loss function for the kth frame is obtained by taking the mean square loss:

$$\mathcal{L}_{3D} = \sum_{i=1}^{k} \left(\omega_k^{gen} - \omega_k^{lip}\right) \tag{15}$$

Where $\omega_k^{gen}$ and $\omega_k^{lip}$ are the lip-movement coefficients and the expression coefficients generated in the wav2lip pre-training, respectively. This loss function expresses the importance of 3D facial features, especially lip-movement features, in face image to face video generation.

At the same time, we calculated the loss function for 3D face projection onto 2D. We represent the facial representation on the 2D projection with the blink signal (which we set as a Gaussian-distributed signal with continuity from 0 to 1).

$$\mathcal{L}_{2D} = \sum_{i=1}^{k} \| \frac{\|\varepsilon_t^{Left1} - \varepsilon_t^{Right1}\|_2 - \|\varepsilon_t^{Left2} - \varepsilon_t^{Right2}\|_2}{2}$$
$$+ \frac{\|\varepsilon_t^{Up1} - \varepsilon_t^{Down1}\|_2 - \|\varepsilon_t^{Up2} - \varepsilon_t^{Down2}\|_2}{2} - z^{style}\|_1 \tag{16}$$

where $z_{style}$ is the blink control signal at frame t, which obeys a Gaussian distribution. $\varepsilon_t^{Left1}$ and $\varepsilon_t^{Right1}$ are used to outline the width region of the left eye, and $\varepsilon_t^{Up1}$ and $\varepsilon_t^{Down1}$ are used to outline the height region of the left eye. The right eye is similar to the left eye. This describes the generation of a continuous blinking performance in a 2D environment.

In addition, we construct a more realistic generated video by head reconstruction. We first pass the generated and the original by applying a mean-square loss between them to compute the reconstruction loss. It is denoted as:

$$\mathcal{L}_{Rebuild} = \frac{1}{k} \sum_{i=1}^{K} \left(\Delta\rho_t' - \Delta\rho_t\right)^2 \tag{17}$$

Moreover, to make the generated faces more plausible in the latent space representation, we encourage the latent space distribution to have similarity to the Gaussian distribution of the mean vector and covariance matrix. Therefore, we define the similarity loss $\mathcal{L}_{kl}$ as the Kullback-Leibler (KL) scatter between the latent spatial distribution and the Gaussian distribution.

Ultimately, the loss function can be expressed as:

$$\mathcal{L}_{loss} = \lambda_{3D}\mathcal{L}_{3D} + \lambda_{2D}\mathcal{L}_{2D} + \lambda_{Rebuild}\mathcal{L}_{Rebuild} + \lambda_{KL}\mathcal{L}_{KL} \tag{18}$$

Where $\lambda_{3D}, \lambda_{2D}, \lambda_{Rebuild}, \lambda_{KL}$ are set to 2, 0.01, 1, 0.7. Based on the above loss function and the method in the main body, the algorithm in this section can be represented by pseudo-code 1.

---

**Algorithm 1** Interaction Module for Dynamic Video Reasonin

---

1: **Input:** Audio sequence $\alpha_{\{1,...,t\}}$, initial face-swapping picture $I_s$, 3DMM parameters $(\bar{F}, r_{id}, r_{ex})$
2: **Output:** Updated video frames with facial head movement
3: Initialize the 3D face shape $F = \bar{F} + \alpha r_{id} + \beta_0 r_{ex}$ using $I_s$ and 3DMM
4: Extract lip motion coefficients $\omega^{lip}$ from pre-trained Wav2Lip model
5: Encode audio features into latent space using ResNeXt-based encoder $\Psi(\alpha_{\{1,...,t\}})$
6: $\mathcal{L}_{3D} \leftarrow 0, \mathcal{L}_{2D} \leftarrow 0, \mathcal{L}_{Rebuild} \leftarrow 0, \mathcal{L}_{KL} \leftarrow 0$
7: **for** $i = 1$ to $t$ **do**
8:     Compute expression coefficients $\beta_i = \Theta(\Psi(\alpha_i), \beta_0 \odot \bar{F})$
9:     Generate head pose parameters $\rho_i = [\mu_i, \nu_i]$ based on conditional VAE
10:     Incorporate style coefficient $z_{style}$ sampled from a Gaussian distribution
11:     Decode the expression and head pose to generate updated frame $I'_i$
12:     $\mathcal{L}_{3D} \leftarrow \mathcal{L}_{3D} + (\omega_i^{gen} - \omega_i^{lip})^2$
13:     $\mathcal{L}_{2D} \leftarrow \mathcal{L}_{2D} + \frac{\|\varepsilon_t^{Left1} - \varepsilon_t^{Right1}\|_2 - \|\varepsilon_t^{Left2} - \varepsilon_t^{Right2}\|_2}{2} + \| \frac{\|\varepsilon_t^{Up1} - \varepsilon_t^{Down1}\|_2 - \|\varepsilon_t^{Up2} - \varepsilon_t^{Down2}\|_2}{2} - z^{style}\|_1$
14:     $\mathcal{L}_{Rebuild} \leftarrow \mathcal{L}_{Rebuild} + (\Delta\rho'_i - \Delta\rho_i)^2$
15:     $\mathcal{L}_{KL} \leftarrow \mathcal{L}_{KL} +$ KL divergence between latent space and Gaussian distribution
16: **end for**
17: $\mathcal{L}_{3D} \leftarrow \mathcal{L}_{3D}/t, \mathcal{L}_{2D} \leftarrow \mathcal{L}_{2D}/t, \mathcal{L}_{Rebuild} \leftarrow \mathcal{L}_{Rebuild}/t, \mathcal{L}_{KL} \leftarrow \mathcal{L}_{KL}/t$
18: $\mathcal{L}_{loss} \leftarrow \lambda_{3D}\mathcal{L}_{3D} + \lambda_{2D}\mathcal{L}_{2D} + \lambda_{Rebuild}\mathcal{L}_{Rebuild} + \lambda_{KL}\mathcal{L}_{KL}$

---

### B.3 VIDEO RE-PREDICTION AND CAUSAL DECISION LOSS.

In our video regeneration, we applied a simple CNN-VIST-CNN nesting approach for video prediction. We set it to MSE loss. Video prediction is defined as: Consider a video sequence $X_{k,T} = \{x_i\}_{k-T+1}^t$ at time $k$, comprising the past $T$ frames. Our objective is to predict the future sequence $X'_{k,T'} = \{x_i\}_k^{t+T'}$ at time $k$, which includes the subsequent $T'$ frames, where each frame $x_i \in \mathbb{R}^{C,H,W}$ is an image characterized by channels $C$, height $H$, and width $W$. Formally, the prediction model is a mapping $\mathcal{F}_\Gamma : X_{k,T} \mapsto X'_{k,T'}$ with learnable parameters $\Gamma$, optimized through:

$$\Gamma' = \arg\min_\Gamma \mathcal{L}\left( \sum_{i=1}^{k-T+1} \mathcal{F}_\Gamma(X_{k,T}), \sum_{i=t}^{t+T'} X'_{k,T'} \right) \tag{19}$$

During this mapping process, it is easy to see that as the number of frames of a given video sequence increases, the more accurate the process is for subsequent predictions. The initial video generation process, on the other hand, relies on the first cover image only, although we chose to use 3D/2D features, head reconstruction potential, etc. to guide the generation of the video. However, inevitably the weight of the first image is very large, almost 1. However, each frame in the time series is relatively independent and has the same weight. Although we believe that the time continuity from the first picture to a video is given, this period of time does not match the original time series spatial sequence, and it is correspondingly difficult to generate a cover video that matches the information that needs to be disseminated in the secret video. Video prediction can be better combined with spatio-temporal information, but few input video sequence frames make it difficult to match the original video spatio-temporal information for a long period. Therefore, we add causal inference to the loss function for decision-making video generation.

For each frame s that requires inference, the cross-entropy loss Ltpred is used to train the classifier for the prediction module:

$$\mathcal{L}_{prediction}^{s} = -\sum_{i=1}^{k} y^k \log(p_t^k) \tag{20}$$

where y is the label vector of the generated frame and $p_t^k$ is the difference between the spatio-temporal dynamics of the kth frame and the original spatio-temporal dynamics. To push the skip policy branch to select a reasonable frame at each inference step, we adopt a simple loss function:

$$\mathcal{L}_{ce}^{k} = \delta_{t-1} - \delta_t \tag{21}$$

where $\delta_t$ is the difference between the frame alteration frame $x_i$ of the best video re-prediction and the other alteration frames with the highest probability. We can simply infer that larger $\delta_t$ indicates more confident and accurate reasoning. Thus, we use this loss function to encourage margins to grow over inference steps, suggesting that at each step, our goal is to select a useful frame to favor our dynamic inference.

We define parameter $\mu$ as a condition of the decision-making program:

$$\mu = \Gamma' - \mathcal{L}_{loss} \tag{22}$$

In the context of a reasoning process, reliable decision-making is imperative if we are to terminate the process at the current reasoning step. However, in the absence of basic fact labels that provide feedback on the viability of exiting the reasoning process, we employ dynamic labels generated using $\delta_t$. The difference in $\delta_t$ across various inference steps allows us to estimate the spatio-temporal information gain obtained from observing additional frames during our dynamic inference process. Given a predefined maximum inference step $T$, the gap between $\delta_t$ (at the current step) and $\delta_T$ (at the final step) serves as an estimate of the potential spatiotemporal information gain if the reasoning is continued to the end. Consequently, this gap can be utilized to determine if the network can terminate reasoning at step $t$. When $\delta_t$ is nearly equal to $\delta_T$, it indicates that the information gained from observing more frames is minimal, allowing us to terminate inference early to reduce computational cost without compromising prediction accuracy. Specifically, at each step $t$, if $\delta_t$ is sufficiently close to $\delta_T$, implying that the estimated residual information gain is negligible, we assign the label $y_{\mathrm{fb}^k}$ to 1, signifying that our model can cease inference at the $t$-th step. Conversely, if the label is set to 0, it indicates the necessity for additional frames. The threshold $\mu > 0$ controls the proximity requirement between $\delta_t$ and $\delta_T$ for the network to exit inference. Therefore, we train the module by minimizing the binary cross-entropy loss:

$$\mathcal{L}_{fb}^{k} = -\left[ y_{fb}^k \log(e_k) + \left(1 - y_{fb}^k\right) \log\left(1 - e_k\right) \right] \tag{23}$$

Therefore, the total loss for the causal decision to choose $x_i$ is:

$$\mathcal{L}_{total} = \mathcal{L}_{fb}^k + \mathcal{L}_{prediction}^s + \lambda_{ce}\mathcal{L}_{ce}^k \tag{24}$$

When

$$\begin{cases} k \leq x_i, \text{Initial Cover Video Generation} \\ k > x_i, \text{Video Re-prediction} \end{cases} \tag{25}$$

Where $\lambda_{ce}$ is the hyperparameter that controls the decision, here set to 3. We accumulate all the inference steps $\mathcal{L}_{total}' = \sum_{i=1}^{k} \mathcal{L}_{total}$ as the final optimization objective.

Based on the above loss function and the method in the main body, the algorithm in this section can be represented by pseudo-code 2.

---

**Algorithm 2** Video Re-prediction and Causal Decision Loss

---

1: **Input:** Video sequence $X_{k,T} = \{x_i\}_{k-T+1}^t$, future sequence length $T'$
2: **Output:** Predicted future sequence $X'_{k,T'} = \{x_i\}_k^{t+T'}$
3: **for** $i = 1$ to $n_i$ **do**
4:     $\Omega_i = \sigma\left(LN\left(C\left(\Omega_{i-1}\right)\right)\right)$
5: **end for**
6: Attention$(Q, K, V) = \text{Softmax}\left(\frac{QK^T}{\sqrt{d_k}}\right)V$
7: MultiHead$(Q, K, V) = \text{Concat}(\text{head}_1, \ldots, \text{head}_h)W^O$
8: head$_i$ = Attention$(QW_i^Q, KW_i^K, VW_i^V)$
9: **for** $i = 1$ to $n_i$ **do**
10:     $\Omega'_i = \sigma\left(LN\left(C\left(\Omega'_{i-1}\right)\right)\right)$
11: **end for**
12: Generate predicted future sequence $X'_{k,T'}$ using the decoder output
13: $\mathcal{L}_{prediction} \leftarrow 0, \mathcal{L}_{ce} \leftarrow 0, \mathcal{L}_{fb} \leftarrow 0$
14: **for** $s = k - T + 1$ to $t + T'$ **do**
15:     $\mathcal{L}_{prediction}^s = -\sum_{i=1}^k y^k \log(p_t^k)$
16:     $\mathcal{L}_{ce}^s = \delta_{t-1} - \delta_t$
17:     **if** $\delta_s \approx \delta_T$ **then**
18:         $y_{\text{fb}^s} \leftarrow 1$ {Early termination is possible}
19:     **else**
20:         $y_{\text{fb}^s} \leftarrow 0$ {More frames are needed}
21:     **end if**
22:     $\mathcal{L}_{fb}^s = -\left[y_{\text{fb}^s}\log\left(e_s\right) + \left(1 - y_{\text{fb}^s}\right)\log\left(1 - e_s\right)\right]$
23:     $\mathcal{L}_{total} \leftarrow \mathcal{L}_{total} + \mathcal{L}_{fb}^s + \mathcal{L}_{prediction}^s + \lambda_{ce}\mathcal{L}_{ce}^s$
24: **end for**
25: $\mathcal{L}_{total} \leftarrow \mathcal{L}_{total}/(t + T' - k + T)$
26: $\Gamma' = \arg\min_\Gamma \mathcal{L}_{total}$

---

### B.4 VIDEO HIDING LOSS.

The loss function for video hiding is mainly a constraint on the reversible neural networks to perform forward-term hiding and backward recovery of the secret video. The forward term hiding is to hide the secret video in the stego video in the cover video. The stego video cannot be recognized as containing the secret video and the generated stego video should be as similar as possible to the cover video. Therefore, we limit $X_{stego}$ to be the same as the cover video $X_{cover}$:

$$\mathcal{L}_{forward} = \|X_{stego} \odot CF - X_{cover} \odot CF\|_2^2 \tag{26}$$

Where CF is the index of the center frame of the video. The output is fused in a time-smoothed manner by means of the frame index. The goal of the backward recovery process is to recover $X_{secret}$ from $X_{stego}$. Thus, we define the loss function as:

$$\mathcal{L}_{backward} = \|\tilde{X}_{secret} \odot CF - X_{secret} \odot CF\|_2^2 + \|\tilde{X}_{cover} \odot CF - X_{cover} \odot CF\|_2^2 \tag{27}$$

Where $\tilde{X}_{secret}$ and $\tilde{X}_{cover}$ denote recovered secret and cover videos.

Finally, we define the objective loss function for video hiding as minimizing the prior hiding loss function and the backward recovery loss function:

$$\mathcal{L}_{VH} = \mathcal{L}_{forward} + \lambda\mathcal{L}_{backward} \tag{28}$$

Where $\lambda$ is the hyperparameter that balances the feed-forward and recovery directions of the reversible neural network and is here set to 2.

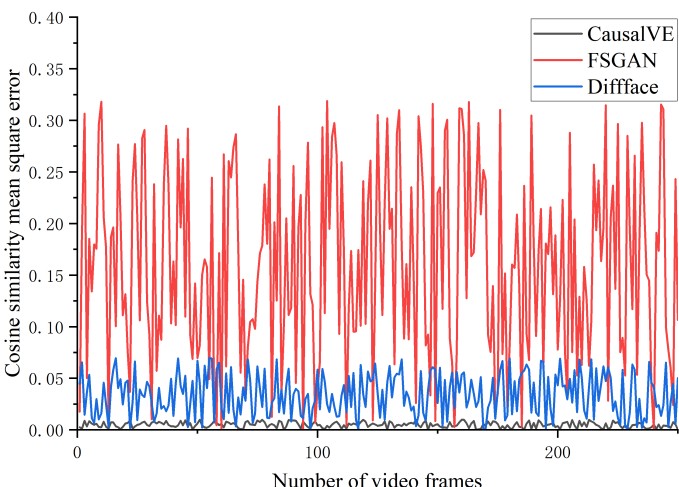

Figure 6: After video hiding by our method and LF-VSN method for videos of the same time sequence, the resulting graphs of comparison of the hidden videos generated by different methods and the same frame of each syllable of a sentence in the original video are taken as images.

## C ABLATION EXPERIMENTS

In order to verify that our method modules are all valid, we study the ablation of our method from the following aspects:

**Question 1:** Can a simple video face-swapping, e.g., using a simple generative model such as GAN or diffusion for video face-swapping, achieve similar results to our model? It can be seen that a complex multi-module model will be far more complex than a simple model during operation. In short, it is whether our modeling study is indispensable.

**Question 2:** How much do temporal and spatial features affect the generation of cover images for publicly distributed videos??

**Question 1 setting.** We still use cosine similarity for comparison. The difference is that in this question, the focus is more on not the mean result of face-swapping, but the stability of face-swapping per frame. We use the mean square deviation of the cosine similarity to represent this.

**Answer to Question 1.** As can be seen from Figure 6, the stability of video face-swapping using GAN is very poor, while the difference between direct video face-swapping using the diffusion model and our method still has a gap in the metrics. This shows that our proposed concept and method are indispensable for privacy preservation in public face video distribution.

**Question 2 setting.** In terms of spatiotemporal feature impact, we predicted video data with different time series lengths by means of speech-video prediction, CNN-ViST-CNN, and causal-time prediction, respectively. In terms of Conv kernel, we investigated the impact of kernel size and hidden dimension on model performance. We used the MSE metric to determine the impact of the Conv kernel.

**Answer to Question 2.** From Table 4, it can be seen that speech video prediction performs comparably to causal temporal prediction for video prediction over short periods. However, speech video prediction is much less effective than causal video prediction in medium and long-term video prediction over 10s. And the CNN-ViST-CNN with direct spatiotemporal evolution is not as effective in short-time face video prediction. This is because speech video prediction makes full use of the face features of the first replacement image in the prediction process, including physical features such as features and expressions. In the actual sequence, the features of a particular frame of the image will keep changing as the time sequence does not grow shorter, and the features of a single image stand for a smaller and smaller proportion of the total time sequence, and other spatiotemporal evolutionary features are needed. The CNN-ViST-CNN that directly performs spatiotemporal evolution does not directly give a lot of attention to a single image, and the spatiotemporal evolution features are not obvious in a short time, which leads to poor prediction results. The trade-off between the two through causal analysis achieves the result of optimal long-time video prediction.

Table 4: **Ablation study in spatiotemporal information similarity**

|     | SpeechVP | CNN-ViST-CNN | CausalVE |
|-----|----------|--------------|----------|
| 1s  | 0.956    | 0.373        | 0.962    |
| 2s  | 0.937    | 0.501        | 0.955    |
| 5s  | 0.842    | 0.474        | 0.936    |
| 10s | 0.771    | 0.435        | 0.942    |
| 20s | 0.682    | 0.417        | 0.938    |
| 30s | 0.594    | 0.409        | 0.943    |

## D  LIMITATIONS

The CausalVE framework involves several advanced techniques such as diffusion models, reversible neural networks, and dynamic causal inference. This complexity may create implementation challenges for practitioners unfamiliar with these methods. Additionally, issues such as different video quality, different lighting conditions, and different types of facial obstructions may affect the performance of the framework. Although the diffusion model can effectively improve the quality of videos, lower video quality still has a great impact on the training of the diffusion model and the generation of new cover videos. Additionally, practical considerations for deployment in real-world applications (such as latency, real-time processing capabilities, and ease of integration) are not considered in this article.

