# OpenReview forum: "CausalVE: Face Video Privacy Encryption via Causal Video Prediction"
_ICLR.cc/2025/Conference — ICLR 2025 Conference Withdrawn Submission_

### Official Review · Reviewer_Yq2Z · 2024-10-19

**Soundness:** 3
**Presentation:** 3
**Contribution:** 2
**Rating:** 5
**Confidence:** 2

**Summary:**

In this paper, the authors propose using video steganography techniques to protect video content shared on public platforms. Specifically, they first perform face swapping on the original video to create a cover video. Then, using a reversible neural network, the original video is embedded into the cover video. Since the cover video is altered, it helps protect the sensitive video information that users wish to safeguard. Through the reversible neural network, the original video can be seamlessly decoded from the cover video, ensuring secure transmission of the video content.

**Strengths:**

- Protecting the original video using video steganography techniques.

- Using symmetric encryption to encode and decode the original video ensures the preservation of information during transmission.

**Weaknesses:**

- Application. The proposed method is positioned as a way to protect user privacy on public platforms. However, this presents an inherent contradiction: if users genuinely wish to protect their privacy, they wouldn’t need to upload videos to a public platform. The only scenario where this might make sense is if users intentionally want to share hidden information through public platforms, which raises potential societal concerns.

- Security. While the use of a reversible neural network ensures video embedding and decoding with minimal loss of quality, the symmetric encryption method itself lacks strong security guarantees.

- Experiments: The paper lacks metrics evaluating video smoothness and realism. The authors are encouraged to use metrics such Fréchet Inception Distance to provide a more detailed assessment of their method’s performance.

**Questions:**

Please see weaknesses.

**Details Of Ethics Concerns:**

The method could potentially allow attackers to safely disseminate harmful content on public platforms through video steganography.

---

### Official Review · Reviewer_Lhus · 2024-10-19

**Soundness:** 2
**Presentation:** 3
**Contribution:** 2
**Rating:** 1
**Confidence:** 4

**Summary:**

This work proposes a new privacy protection framework for face video. The framework includes three modules: face swapping, video prediction, and video steganography, which realizes the high visual quality of the cover video and the strong undetectability of the secret video.

I am sorry that this work should be rejected because it has many weaknesses.

**Strengths:**

1. A new understandable framework to protect face privacy in video.

2. The structure of the writing is clear

**Weaknesses:**

1. Motivation of the Work is Ambiguous: The rationale behind the need for reversibility is unclear. Specifically, I do not see the significance of restoring the original video. The primary purpose of privacy protection is to eliminate sensitive information. In particular, "irreversibility" would more effectively enhance the strength of privacy protection. The authors need to clarify the scenarios in which reversibility is applicable.

2. Implementation Approach is Unreasonable: The authors propose hiding a secret video within a cover video to protect facial privacy. However, I find this approach difficult to accept. This is because the secret video and the cover video share highly similar content (i.e., attributes other than identity), making it unnecessary to hide the secret video directly. In other words, since the only difference between the secret and cover videos is the identity, would it not be more feasible to simply hide the identity instead?

3. Technical Innovation is Weak: The authors have integrated techniques such as face swapping, video prediction, and video steganography to construct the CausalVE framework, which makes it hard to identify any significant technical innovation in this work. The three contributions mentioned in the "primary contributions" section are all achievable by existing methods; this paper merely applies these techniques to facial privacy protection.

4. Insufficient Experimental Evaluation: (1) Incorrect Comparison Trials: This work aims to protect facial privacy, and it should be compared with existing facial privacy protection methods rather than video steganography, like [1] or [2]. (2) Robustness Evaluation: Videos encoded in different formats may lose some information. Can this framework still achieve reversibility under such conditions?

[1]The UU-Net_ Reversible Face De-Identification for Visual Surveillance Video Footage.
[2]IdentityMask_Deep_Motion_Flow_Guided_Reversible_Face_Video_De-identification

5. Practicality of the Work is Poor: The proposed framework incorporates various technologies and losses, making it difficult for users to understand. Additionally, the use of time-consuming techniques such as diffusion models results in high energy consumption and latency for CausalVE, complicating its integration into practical applications.

**Questions:**

1. Why does the framework need reversibility？

2. Why not just hide the identity？

3. What is the technical innovation of the work? Please explain how each part of the work (face swapping, video prediction, video steganography) is novel relative to existing techniques.

---

### Official Review · Reviewer_JgqY · 2024-10-31

**Soundness:** 2
**Presentation:** 1
**Contribution:** 2
**Rating:** 3
**Confidence:** 4

**Summary:**

This paper addresses bioprivacy concerns raised by advanced facial recognition and recommendation systems that lack sufficient privacy protections, especially with the spread of video and live-streaming platforms. Current methods to prevent biometric information leakage often compromise security by either distorting interaction data or leaving identifiable features vulnerable. To address these gaps, the proposed neural network framework, CausalVE, uses a diffusion model for face-swapping guided by facial features and a reversible neural network for securely embedding the original video content. Extensive experiments show that CausalVE effectively secures public video dissemination and surpasses current methods in both quality and privacy protection.

According to the author's statement, this research is indeed a hot field. However, there are some problems in the manuscript that have to attract the attention of the authors.

**Strengths:**

- The CausalVE framework uses causal reasoning to guide the video prediction process, producing cover videos that are both visually convincing and capable of securely carrying hidden information.

- This framework leverages a reversible neural network, allowing the original video to be concealed within a pseudo-video and accurately recovered using a key, thereby safeguarding personal data while enabling secure public distribution.

- CausalVE incorporates a hybrid diffusion model that uses identity features and controlled noise processes to create cover face videos, effectively concealing real identity information while preserving the authenticity and expressiveness of facial features.

**Weaknesses:**

**Some major comments:**

- The manuscript lacks some visual results display and qualitative evaluation.

- The framework proposed in this manuscript integrates multiple tasks, resulting in incomplete introduction of each task.

**Some other minor comments:**

- The author's summary of innovation is too confusing. I need to spend time reading the full text to understand it. The author still needs to strengthen his writing of the manuscript.

- The drawings in this manuscript are too rough, and the author still needs to improve in this aspect.

- There are too many redundant descriptions in the manuscript.

- There is no description of relevant work in the main text. Although the author provides relevant descriptions in the supplementary materials, it is only a list of some work.

**Questions:**

- The author needs to provide a description of the time-consuming of the algorithm.

- The author needs to increase the discussion of generalization.

---

### Official Review · Reviewer_WRXn · 2024-11-01

**Soundness:** 2
**Presentation:** 1
**Contribution:** 2
**Rating:** 5
**Confidence:** 5

**Summary:**

The tremendous success of face recognition models, sometimes witness the illegal access of private information from facial images. Existing privacy-preserving techniques leave sensitive information that an attacker can easily infer for malicious purposes. In response, the author presents a CASUAL-VE approach of face swapping where the guidance of that has been obtained through the diffusion models. The experiments are performed using three dataset to showcase the effectiveness of the proposed approach using multiple evaluation metrics.

**Strengths:**

- The protection of privacy is a genuine concern and efforts towards that are highly needed.
- Although not novel, still the use of a cover image through face guidance is interesting.

**Weaknesses:**

- One of the primary weaknesses of the paper is its editorial limitation. The paper is hard to read and follow. For example, a significant amount of information is missing or not adequately presented. For instance, in line 094, what physical information has been used? What is the role of a pseudo-video (line 100)? At line 100, what form of frequency is used to divide frames?

- The motivation for using the diffusion model is not clear. The field of steganography is not new and several research works have used the image-hiding technique to generate adversarial examples.

[1] Zhang Y, Zhang W, Chen K, Liu J, Liu Y, Yu N. Adversarial examples against deep neural network-based steganalysis. In Proceedings of the 6th ACM Workshop on information hiding and multimedia security 2018 Jun 14 (pp. 67-72).

[2] Agarwal A, Ratha N, Vatsa M, Singh R. Crafting adversarial perturbations via transformed image component swapping. IEEE Transactions on Image Processing. 2022 Sep 12;31:7338-49.

[3] Din SU, Akhtar N, Younis S, Shafait F, Mansoor A, Shafique M. Steganographic universal adversarial perturbations. Pattern Recognition Letters. 2020 Jul 1;135:146-52.

- Apart from image-hiding techniques, literature extensively uses adversarial noises for privacy-preserving face recognition where the authors mask the sensitive information. The authors must compare with these existing works and along with that perform experiments showcasing the impact on face recognition and soft attribute prediction to better reflect the solution of privacy concerns.

[4] Chhabra S, Singh R, Vatsa M, Gupta G. Anonymizing k-facial attributes via adversarial perturbations. In Proceedings of the 27th International Joint Conference on Artificial Intelligence 2018 Jul 13 (pp. 656-662).

- The ablation studies concerning frequency decomposition techniques such as FFT, DCT, and DWT can be compared. The role of a cover image can be effectively studied. How the change in a cover image can affect privacy?

- The authors have used several metrics, but which one metric is most appropriate is not clear. Further, a statistical test is needed to showcase whether the proposed values are significantly better than the existing values, especially from LF-VSN.

- The paper utilizes a technique proposed in 2014 for steganalysis (line 366). I suggest the use of any recent and state-of-the-art model.

**Questions:**

- The motivation for using the diffusion model is not clear. The field of steganography is not new and several research works have used the image-hiding technique to generate adversarial examples.

- Apart from image-hiding techniques, literature extensively uses adversarial noises for privacy-preserving face recognition where the authors mask the sensitive information. The authors must compare with these existing works and along with that perform experiments showcasing the impact on face recognition and soft attribute prediction to better reflect the solution of privacy concerns.

- The ablation studies concerning frequency decomposition techniques such as FFT, DCT, and DWT can be compared. The role of a cover image can be effectively studied. How the change in a cover image can affect privacy?

- The authors have used several metrics, but which one metric is most appropriate is not clear. Further, a statistical test is needed to showcase whether the proposed values are significantly better than the existing values, especially from LF-VSN.

---

### Official Review · Reviewer_j9sv · 2024-11-04

**Soundness:** 3
**Presentation:** 2
**Contribution:** 3
**Rating:** 5
**Confidence:** 4

**Summary:**

The paper proposes CausalVE, a face video privacy protection framework that combines (1) a diffusion model for face swapping with facial guidance, (2) a video prediction method that uses speech and spatiotemporal visual features of the secret facial video to generate a realistic cover video, and (3) a reversible neural network to embed the secret video within the cover video. This reversible neural network enables retrieval of the original video using a specific key. By balancing data concealment with coherent video output, CausalVE aims to enhance privacy protection beyond traditional methods. Evaluation results indicate that CausalVE effectively safeguards private information, outperforming baseline methods.

**Strengths:**

1. CausalVE combines causal reasoning, reversible neural networks, and hybrid diffusion models to achieve high-fidelity face swapping and robust privacy preservation.

2. The framework offers privacy protection without compromising video quality, enabling natural and realistic facial video transformations.

3. By embedding the original video within the cover video, CausalVE maintains a balance between privacy and the potential for legitimate retrieval, thanks to its reversible neural network.

**Weaknesses:**

1. The paper could mislead readers into believing that the entire video frame is processed and hidden rather than just the facial region. Since the facial region occupies only part of a frame, the data requirements for concealing and generating only the face differ substantially from handling the entire frame. Clarifying this distinction early on would improve readability and prevent misunderstandings.

2. Due to the potential for confusion about the processing scope, it’s unclear whether the paper makes fair comparisons with other methods. For example, did it use the same cropped facial region as a hidden element for fair benchmarking with other baselines?

3. The paper's references to 3D-based deepfake techniques are somewhat outdated. Incorporating recent literature such as those in [*] would provide a more current perspective on related methodologies.

4. The framework’s computational demands are significant, which could limit accessibility and scalability for resource-limited settings.

[*] Pei, Gan, Jiangning Zhang, Menghan Hu, Zhenyu Zhang, Chengjie Wang, Yunsheng Wu, Guangtao Zhai, Jian Yang, Chunhua Shen, and Dacheng Tao. "Deepfake generation and detection: A benchmark and survey." arXiv preprint arXiv:2403.17881 (2024).

**Questions:**

Please address the concerns outlined in the weaknesses section.

---

### Author Response · Authors · 2024-11-23

We are very grateful to all reviewers for their valuable suggestions. We are working hard to complete the additional experiments requested by each reviewer and to respond to each reviewer's comments as soon as possible. Thanks!

---

### Note · Authors · 2024-12-04

I have read and agree with the venue's withdrawal policy on behalf of myself and my co-authors.